# The influence of observation sequence features on the performance of the Bayesian hidden Markov model: A Monte Carlo simulation study

Jan-Willem Simons[1]*, Bart-Jan Boverhof[2], Emmeke Aarts[3]*

**1** Department of Sociology, Utrecht University, Utrecht, The Netherlands, **2** Erasmus School of Health Policy & Management, Erasmus University Rotterdam, Rotterdam, The Netherlands, **3** Department of Methodology and Statistics, Utrecht University, Utrecht, The Netherlands

\* j.g.simons@uu.nl (JWS); e.aarts@uu.nl (EA)

**Data Availability Statement:** R code to reproduce and analyze the simulation study and the empirical application along with the data are openly available

## Abstract

The hidden Markov model is a popular modeling strategy for describing and explaining latent process dynamics. There is a lack of information on the estimation performance of the Bayesian hidden Markov model when applied to categorical, one-level data. We conducted a simulation study to assess the effect of the 1) number of observations (250—8.000), 2) number of levels in the categorical outcome variable (3—7), and 3) state distinctiveness and state separation in the emission distribution (low, medium, high) on the performance of the Bayesian hidden Markov model. Performance is quantified in terms of convergence, accuracy, precision, and coverage. Convergence is generally achieved throughout. Accuracy, precision, and coverage increase with a higher number of observations and an increased level of state distinctiveness, and to a lesser extent with an increased level of state separation. The number of categorical levels only marginally influences performance. A minimum of 1.000 observations is recommended to ensure adequate model performance.

## Introduction

The hidden Markov model (HMM) is a popular modeling strategy for describing and explicating latent process dynamics [1]. It can be defined as a dynamic Bayesian network that models a system of unobserved states as a first-order Markov process, where the unobserved process is inferred from a second observed process [2]. The HMM is a flexible modeling framework that enables researchers to retrieve and study a wide array of latent process phenomena. Examples include but are not limited to biological sequence analysis [3], brain magnetic resonance image segmentation [4], human action [5] and shape classification [6], speech [7] and handwriting recognition [8], precipitation [9], and stock market forecasting [10].

One of the key tasks of the HMM is to estimate its parameters so that they maximize the probability of an observation sequence given the HMM [11]. Obtaining a high quality solution to this optimization task is key to most applied research because it enables researchers to

on Zenodo at https://doi.org/10.5281/zenodo.11186671.

**Funding:** The author(s) received no specific funding for this work.

**Competing interests:** The authors have declared that no competing interests exist.

model real-world phenomena. The features of the observation sequence are a crucial determinant of the quality of this solution [11]. In this simulation study, we assess if and how three central features of categorical observation sequences affect the performance of the Bayesian HMM: (1) the number of observations comprising the sequence, (2) the number of levels in the categorical outcome variable, and (3) state distinctiveness and separation. Our focus on categorical longitudinal data is driven by the increasing availability of such data in the social and behavioral sciences. Technological advancements have improved the efficiency and affordability of gathering high-resolution data from individuals and groups [12–16]. Much of this data is furthermore categorical in nature, e.g., reflect social actions like drinking, speaking, and laughing during free-standing conversations and speed dating [13] or interactive behaviours in group therapies for people with schizophrenia, like asking questions, giving advice, and using humor [16].

However, there currently is a lack of information on how the features of such categorical observation sequences affect Bayesian solutions to the optimization task. Recently, for a specific extension of the Bayesian HMM, simulation studies have investigated how the number of observations and state distinctiveness and separation affect parameter estimation [17–19]. These studies relate to the multilevel Bayesian HMM [20], a model designed to simultaneously summarize the sequences of multiple individuals or animals. In the multilevel HMM, individual-level random effects are included to accommodate (unexplained) heterogeneity in model parameters while estimating one overall HMM. It was shown that in multilevel HMMs fitted on categorical data, the number of individuals and the number of observations in the sequence per individual largely compensate for each other in supporting model performance [19]. In the case of uni-variate data, a minimum of 800 observations on five individuals were required to obtain adequate model performance in a data set where the states are distinctive and well separated [19]. We define 'one-level data' as sequence data collected at a single level of measurement for each individual, in contrast to multi-level data, where multiple sequences of data are collected for each individual and nested within these individuals in the analysis. Additionally, 'uni-variate data' refers to data where only a single outcome variable is measured at each point in time, rather than multiple outcomes. To our knowledge, no simulation studies exist for the basic, one-level Bayesian HMM.

As such, for one-level data, it is unknown what constitutes a minimum sample size to achieve adequate model performance and how this minimum sample size varies with the number of categorical levels and state distinctiveness and separation. The aim of this study is to address this gap and add to a growing body of data guidelines on the Bayesian HMM [19]. Specifically, we explore the effect of the number of observations on the model performance of the Bayesian HMM with data varying in the number of categorical levels and state distinctiveness and separation. We address the following research questions: *RQ1: How do the number of observations affect the estimation performance of the Bayesian HMM? RQ2: How does the estimation performance of the Bayesian HMM vary with the number of categorical levels? RQ3: How do state distinctiveness and separation influence the performance of the Bayesian HMM*?

The number of observations is a design feature and is generally under the control of the researcher. We expect that an increase in the number of observations that are stochastically linked to the latent process of interest will result in an increase in the amount of available information about that latent process. On that basis, we hypothesize that an increase in the number of observations in the observation sequence will result in an increase in the performance of the Bayesian HMM. With respect to the number of categorical levels, an increase in the number of categorical levels will result in an increase in the total number of to be estimated model parameters. We hypothesize that this increase in the complexity of the to-be-estimated Bayesian HMM will result in a decrease in its performance.

State distinctiveness and separation are inherent to the data-generating process and are generally not under the researcher's control. State distinctiveness refers to the degree to which a hidden state is meaningfully associated with one or more categories (i.e., key indicators) without being obscured by random noise. High state distinctiveness is evident when one or a few categories display a high state-dependent emission probability while all other categories exhibit near-zero emission probabilities. Low state distinctiveness relates to high noise in the emission distribution, reflected by a reduced difference between the emission probabilities of key indicators and noise categories. State separation refers to the degree to which state-dependent emission distributions overlap. In categorical data, high state separation is evident when a key indicator is only related to one hidden state. In contrast, given lower state separation, key indicators are related to multiple hidden states instead. As higher state distinctiveness and state separation result in more salient states, we hypothesize that higher state distinctiveness and separation will increase the performance of the Bayesian HMM.

## Materials and methods

### The hidden Markov model

The hidden Markov model (HMM) assumes that 1) the distribution of the observations $O$ observed at time point $t \in \{1, 2, \ldots, T\}$ depends on a sequence of hidden states $S_t = i$, $i \in \{1, 2, \ldots, m\}$ and that 2) the hidden state sequence follows a first-order Markov process [11]. That is, the probability of switching from state $i$ at time point $t$ to state $j$ at $t + 1$ only depends on the departing state $i$ at time point $t$. Fig 1 provides a depiction of the temporal evolution of the HMM.

Given these assumptions, the HMM can be defined on the basis of three sets of parameters. The initial state probabilities $\pi_i$ denote the probability that the first state in the chain, $S_1$, is $i$:

$$\pi_i = Pr(S_1 = i) \text{ with } \sum_i^{\pi_i} = 1. \tag{1}$$

The transition probability matrix $\Gamma$ with transition probabilities $\gamma_{ij}$ denote the probability of switching from state $i$ at time $t$ to state $j$ at time $t + 1$:

$$\gamma_{ij} = Pr(S_{t+1} = j | S_t = i) \text{ with } \sum_j^{\gamma_{ij}} = 1. \tag{2}$$

For each point in time $t$, we have one hidden state that generates one observed outcome for that time point $t$. In this paper we focus on categorical data, as such the state-dependent

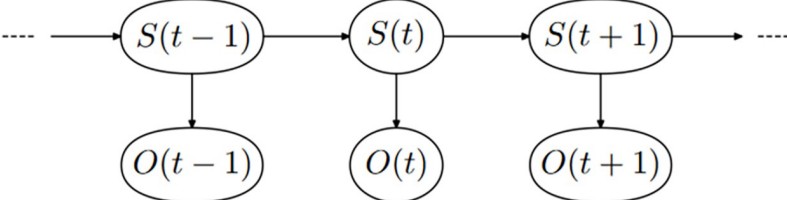

**Fig 1. Directed acyclic graph of the hidden Markov model.** Each hidden state $S$ over the time points $t$, depicted in the top row, depends only on the state at the previous time point. The observed data $O$, depicted in the bottom row, depends only on the value of the current latent state $S$.

emission distribution denotes the probability of observing $O_t$ given $S_t$:

$$Pr(O_t = o | S_t = i) \sim Cat(\theta_i) \tag{3}$$

for the observed categorical levels $o = 1, 2, \ldots, q$ and where $\theta_i = (\theta_{i_1}, \theta_{i_2}, \ldots, \theta_{i_q})$ is a vector of probabilities for each state $S = i, \ldots, m$ with $\sum \theta_i = 1$. We finally assume that all parameters in the HMM are independent of $t$, i.e., we assume a time-homogeneous model.

## The Gibbs sampler

The Gibbs sampler is a Markov chain Monte Carlo (MCMC) algorithm that can be used to estimate the parameters of the HMM by sampling from the joint posterior distribution of the parameter estimates and the hidden state sequence given a set of hyper-parameters and the observation sequence [21]:

$$\Pr\left((S_t), \Gamma_i, \theta_i \mid (O_t)\right) \propto \Pr\left((O_t)|(S_t), \theta_i\right) \Pr\left((S_t) \mid \Gamma_i\right) \Pr\left(\Gamma_i \mid a_{10}\right) \Pr\left(\theta_i \mid a_{20}\right). \tag{4}$$

The Gibbs sampler iteratively samples from the appropriate conditional posterior distributions of $S_t$, $\Gamma_i$ and $\theta_i$, given the remaining parameters in the model. The conditional distributions are constructed from a prior distribution and the likelihood function of the observation sequence. Since both the transitions from state $i$ at time point $t$ to any of the other states at time point $t + 1$ and the observed outcomes within state $i$ follow a categorical distribution, the Dirichlet distribution is a convenient conjugate prior distribution on these parameters. Here, we assume that the rows of $\Gamma$ and the state-dependent probabilities $\theta_i$ are independent. As such.

$$S_{t=2,\ldots,T} \sim \Gamma_{S_{t-1}} \text{ with } \Gamma_i \sim \text{Dir}(a_{10}) \tag{5}$$

and

$$O_{t=1,\ldots,T} \sim \theta_{S_t} \text{ with } \theta_i \sim \text{Dir}(a_{20}). \tag{6}$$

The hyper-parameter $a_{10}$ of the prior Dirichlet distribution on $\Gamma_i$ is a vector with length equal to the number of states $m$, and the hyper-parameter $a_{20}$ of the prior Dirichlet distribution on $\theta_i$ is a vector with length equal to the number of categorical levels $q$. In this study we utilize uninformative priors, as such the hyper-parameter values are fixed to a vector of 1's for both $a_{10}$ and $a_{20}$. As the initial probabilities of the states $\pi_i$ are assumed to coincide with the stationary distribution of $\Gamma$, $\pi_i$ is not estimated freely.

Given the conditional distributions, the Gibbs sampler alternates between the following two steps: first the hidden state sequence $S_1, S_2, \ldots, S_T$ is sampled, given, the observation sequence $O_1, O_2, \ldots, O_T$, and the current values of the parameters $\Gamma$ and $\theta_i$. Forward-backward recursions are used to sample from the hidden state sequence of the HMM [22]. This procedure first obtains the forward probabilities

$$\alpha_t(i) = \Pr\left(O_1 = o_1, O_2 = o_2, \ldots, O_t = o_t, S_t = i\right)$$

after which the hidden state sequence is sampled in a backward run using the corresponding forward probabilities $\alpha_{T:1}$. Second, $\Gamma_i$ and $\theta_i$ are subsequently updated by sampling them conditional on the sampled hidden state sequence $S_1, S_2, \ldots, S_T$ and the observation sequence $O_1, O_2, \ldots, O_T$.

## Methods

The aim of the Monte Carlo simulation study was to empirically assess the performance of the Bayesian hidden Markov model (HMM), varying the number of observations, level of state

distinctiveness and separation, and number of categorical levels. A description of the levels used for each of the simulation factors follows below. A fractional factorial design was used to save computational costs, as detailed below.

**Number of observations and categorical levels.** We base the ranges of the number of observations $n$ and the number of categorical levels $q$ on a sample of studies from a systematic review on the HMM and its applications [1]. We include in this sample all the studies in this review that use a basic frequentist or Bayesian HMM for parameter estimation. Within the resulting sample of studies, the number of observations of the observation sequence ranges from four to 1.2 million [23–51]. However, this distribution has a strong positive skew. To reflect a range that will be relevant to most researchers, we considered six levels for the number of observations $n \in \{250, 500, 1000, 2000, 4000, 8000\}$. The varying number of observations are crossed with the full range of levels of the remaining factors varied.

The number of categorical levels within the systematic review ranges from three to 100. Again, this distribution has a strong positive skew. On that basis, we considered three levels for the number of categorical levels $q \in \{3, 5, 7\}$. In addition, by varying the categorical emission probabilities over the rows of the emission probability matrix, we tentatively assess the effect of a state having one, two, or three indicators.

**State distinctiveness and separation.** The state distinctiveness and state separation features are both assigned three levels: "High", "Moderate", and "Low". Within our simulation study, the categorical emission distributions are constructed such that one to three categories have a high emission probability within a given state $S$. We refer to these as the indicator categories. Non-indicator categories we refer to as noise categories. Given that all emission probabilities within a state sum to one, larger probabilities for the noise categories inevitably result in lower probabilities for the indicator categories and hence lower state distinctiveness. As such, state distinctiveness can be seen as a signal to noise ratio (SNR) of the indicator category: $SNR = P_{indicator}/P_{noise}$, with higher values representing higher state distinctiveness. Depending on the number of categorical levels and number of indicator variables within a state, the SNR ranges between 8.00 and 46.00 for High state distinctiveness, between 4.75 and 8.50 for Moderate state distinctiveness, and between 2.07 and 4.00 for Low state distinctiveness. Over the levels of state distinctiveness, the SNR values steadily increase within any given number of categorical levels. Only within the scenario of 5 categorical levels are all levels of the factor state distinctiveness applied. For the remaining levels of the factor categorical levels, only high and low state distinctiveness is used.

The amount of state separation is quantified using the summed pairwise Kullback–Leibler divergence $D_{KL}$, with higher values denoting a larger difference, i.e., state separation. In a model with $m = 3$ states, the pairwise comparisons refer to the $D_{KL}$ of state 1 versus 2, state 1 versus 3, and state 2 versus 3, with the summed pairwise $D_{KL}$ referring to the summation over these three comparisons. In the High state separation scenario, each category is a unique indicator for one hidden state only. In the moderate state separation scenario, each hidden state is composed of one or two categories that are a unique indicator to that hidden state only, plus a category that is shared as an indicator over two states. Here, the probability of the unique indicator is higher compared to the shared indicator. In the low state separation scenario, the probabilities of the unique and shared indicators are equal. This results in a summed pairwise $D_{KL}$ for the High, Moderate, and Low state separation scenarios of 6.65, 4.44 and 3.88, respectively. Only within the scenario of 5 categorical levels, varying levels of state separation are examined. To keep this study concise, the levels of state distinctiveness and state separation are not crossed. An overview of the used emission probability matrices is provided in Table 1.

**Data generation and model fitting.** We generated 500 sequences for each simulation scenario. To determine the amount of hidden states, we screened recent articles (published after

**Table 1. Emission probabilities within each of the simulation scenarios varying the factors number of categorical levels, state distinctiveness and state separation.**

| | Number of categorical levels | | |
|---|---|---|---|
| | **Three** | **Five** | **Seven** |
| **State distinctiveness** | | | |
| High | $\begin{pmatrix} 0.80 & 0.10 & 0.10 \\ 0.10 & 0.80 & 0.10 \\ 0.10 & 0.10 & 0.80 \end{pmatrix}$ | $\begin{pmatrix} 0.92 & 0.02 & 0.02 & 0.02 & 0.02 \\ 0.02 & 0.47 & 0.02 & 0.47 & 0.02 \\ 0.02 & 0.02 & 0.47 & 0.02 & 0.47 \end{pmatrix}$ | $\begin{pmatrix} 0.76 & 0.04 & 0.04 & 0.04 & 0.04 & 0.04 & 0.04 \\ 0.02 & 0.45 & 0.02 & 0.45 & 0.02 & 0.02 & 0.02 \\ 0.01 & 0.02 & 0.31 & 0.02 & 0.31 & 0.02 & 0.31 \end{pmatrix}$ |
| Moderate | Excluded | $\begin{pmatrix} 0.68 & 0.08 & 0.08 & 0.08 & 0.08 \\ 0.08 & 0.38 & 0.08 & 0.38 & 0.08 \\ 0.08 & 0.08 & 0.38 & 0.08 & 0.38 \end{pmatrix}$ | Excluded |
| Low | $\begin{pmatrix} 0.52 & 0.24 & 0.24 \\ 0.24 & 0.52 & 0.24 \\ 0.24 & 0.24 & 0.52 \end{pmatrix}$ | $\begin{pmatrix} 0.44 & 0.14 & 0.14 & 0.14 & 0.14 \\ 0.14 & 0.29 & 0.14 & 0.29 & 0.14 \\ 0.14 & 0.14 & 0.29 & 0.14 & 0.29 \end{pmatrix}$ | $\begin{pmatrix} 0.40 & 0.10 & 0.10 & 0.10 & 0.10 & 0.10 & 0.10 \\ 0.10 & 0.25 & 0.10 & 0.25 & 0.10 & 0.10 & 0.10 \\ 0.08 & 0.08 & 0.22 & 0.08 & 0.22 & 0.09 & 0.22 \end{pmatrix}$ |
| **State separation** | | | |
| High | Excluded | $\begin{pmatrix} 0.84 & 0.04 & 0.04 & 0.04 & 0.04 \\ 0.04 & 0.44 & 0.04 & 0.44 & 0.04 \\ 0.04 & 0.04 & 0.44 & 0.04 & 0.44 \end{pmatrix}$ | Excluded |
| Moderate | Excluded | $\begin{pmatrix} 0.59 & 0.29 & 0.04 & 0.04 & 0.04 \\ 0.04 & 0.29 & 0.59 & 0.04 & 0.04 \\ 0.04 & 0.04 & 0.20 & 0.36 & 0.36 \end{pmatrix}$ | Excluded |
| Low | Excluded | $\begin{pmatrix} 0.44 & 0.04 & 0.04 & 0.04 & 0.04 \\ 0.04 & 0.44 & 0.04 & 0.44 & 0.04 \\ 0.04 & 0.04 & 0.30 & 0.31 & 0.31 \end{pmatrix}$ | Excluded |

Note that each of the given scenarios was investigated using the full range for the factor number of observations.

2000) which used either basic frequentist or Bayesian HMMs for parameter estimation [23, 27, 28, 32–36, 38, 43, 44, 50] as reported in [1]. This procedure revealed that most studies analyzed between 2 and 15 hidden states, with the majority focusing on the 3 to 5 range. On that basis, we generated data by setting the number of states to $m$ = 3. The values of the transition probability matrix are set to 0.80 on the diagonal and 0.10 on the off-diagonal. Simulating data and fitting of models in the statistical software R [52] using the R CRAN package mHMMbayes [53]. Corresponding R code to reproduce and analyze the simulation study can be found in [54]. In the mHMMbayes package, one-level HMMs are fitted using a Gibbs Markov chain Monte Carlo (MCMC) as detailed in the the section 'The Gibbs sampler'. A total of 5000 MCMC iterations were used, with a burn-in of 1000 for each estimation run. Informative starting values were specified for the transition and emission probabilities to prevent label switching. The reader is referred to S1 Appendix for an overview.

To check convergence of the model parameters, two additional chains were fitted using different starting values for the first ten generated sequences for each scenario in the simulation design. The potential scale reduction factor (PSRF) is subsequently calculated over the chains of each replication. The PSRF measures approximate convergence as the ratio between a mixture of the within-chain and cross-chain variances and the within-chain variance [55]. If the chains have converged, the PSRF will be close to one. Values larger than 1.20 indicate a lack of convergence [56]. To summarize convergence across the parameters of one fitted model, a PSRF-based criterium is used for assessing convergence: parameter convergence is achieved when the PSRF is between one and 1.20 *all parameters* across each of the ten replications. Additionally, the chains are overlaid by way of trace plots to inspect mixing, diagnose potential label switching, and contextualize findings with respect to the PSRF criterium.

The performance of the Bayesian HMM is assessed using the following three criteria: the mean relative mean bias, precision (empirical model standard error), and coverage of the 95% credible intervals (CrI). We additionally provide the mean absolute mean bias and bias-corrected coverage of the 95% CrI in S1 Fig. For reasons of brevity, the mean precision is also provided in S1 Fig. We opt for the relative mean bias over the mean absolute bias for diagnosing accuracy as an 'acceptable' bias range can be specified. However, for very small ground truth parameter values, the relative mean bias will be inflated. In such cases, we will use the mean absolute bias to contextualize the findings. To keep the results section concise, results for the average absolute and relative bias, empirical model standard error, and coverage are summarized across conceptually equivalent transition and emission probability parameters. For the transition probabilities, we distinguish between the self and the state-to-state transition probabilities. For the emission probabilities, we distinguish between the probabilities relating to the indicator and noise categories. The standard deviation—as the degree to which the mean of a metric varies across conceptually equivalent groups—is presented as a band for all metrics, except for the width of the 95% CI, for which it is presented as an error bar.

## Results

### Convergence

Table 2 shows the proportion of transition and emission parameters that reached adequate convergence within each cell of the simulation design. Full or near-to-full convergence was generally achieved across all transition and emission parameters on the high and moderate levels of the state distinctiveness and all levels of state separation, irrespective of the number of categorical levels. For the slightly more challenging scenarios, e.g., moderate state distinctiveness and moderate and low state separation, 250 and 500 appeared to be insufficient to

**Table 2. Proportion of transition and emission probability parameters that have converged on the basis of the potential scale reduction factor criterion.**

| | | Transition parameters | | | | | | Emission parameters | | | | | |
|---|---|---|---|---|---|---|---|---|---|---|---|---|---|
| | | Number of observations | | | | | | Number of observations | | | | | |
| | | 250 | 500 | 1000 | 2000 | 4000 | 8000 | 250 | 500 | 1000 | 2000 | 4000 | 8000 |
| State distinctiveness | Number of categorical levels | | | | | | | | | | | | |
| High | Three | 0.67 | 1 | 1 | 1 | 1 | 1 | 0 | 1 | 1 | 1 | 1 | 1 |
| | Five | 1 | 1 | 1 | 1 | 1 | 1 | 1 | 1 | 0.93 | 0.93 | 1 | 1 |
| | Five (Non-uniform) | 1 | 1 | 1 | 1 | 1 | 1 | 1 | 1 | 0.93 | 1 | 1 | 0.93 |
| | Seven | 1 | 1 | 1 | 1 | 1 | 1 | 1 | 1 | 1 | 0.95 | 0.95 | 0.86 |
| Moderate | Five | 0.33 | 0.11 | 1 | 1 | 1 | 1 | 0 | 0 | 1 | 0.93 | 1 | 1 |
| Low | Three | 1 | 0.67 | 0.33 | 0 | 0 | 0 | 0.78 | 0.22 | 0 | 0 | 0 | 0 |
| | Five | 0.89 | 0 | 0 | 0 | 0 | 0 | 0.73 | 0.13 | 0 | 0 | 0 | 0 |
| | Seven | 0.89 | 0.11 | 0 | 0 | 0 | 0 | 0.81 | 0.29 | 0.05 | 0 | 0.19 | 0.19 |
| State separation | | | | | | | | | | | | | |
| High | Five | 1 | 1 | 1 | 1 | 1 | 1 | 1 | 1 | 1 | 1 | 1 | 1 |
| Moderate | Five | 0 | 0.11 | 1 | 1 | 1 | 1 | 0 | 0 | 1 | 0.93 | 1 | 1 |
| Low | Five | 0 | 0 | 1 | 1 | 1 | 0.67 | 0 | 0 | 0.93 | 0.93 | 0.93 | 0.87 |

Parameter convergence is achieved when the PSRF is between one and 1.20 for *all parameters* across each of the replications.

achieve adequate convergence. The proportion of converged parameters increased with increasing number of observations.

With low state distinctiveness, adequate convergence appears to only be achieved with 250 observations. However, inspection of trace plots revealed that chains do not converge and instead show drift and high variability. As a result, the within-chain variance is high compared to the between-chain variance, producing a corresponding PSRF value that is close to one. When the number of observations increases, the chains start to approach the appropriate stationary distribution, lowering the within-chain variance and the PSRF value. We note that the simulation scenarios that suffer from sub-par convergence should be interpreted with a measure of caution in the following sections.

## Accuracy

For the transition probabilities, the mean relative bias generally reached the acceptable level of $< |0.1|$ for high and moderate state distinctiveness and all levels of state separation, except when the number of observations = 250, see Fig 2. For the emission probabilities, the mean relative bias generally reached the acceptable level of $< |0.1|$ for high and moderate state distinctiveness and all levels of state separation when the number of observations > 250 (high and moderate state distinctiveness and separation) or > 500 (low state separation), see Fig 3. For both transition and emission probabilities, all scenarios showed that relative bias tends to zero with an increasing number of observations. In the case of low state distinctiveness, relative bias only managed to touch the range of acceptable values at number of observations = 8.000 for transition probabilities and number of observations = 4.000 for emission probabilities. The number of categorical levels only marginally influenced relative bias, with 7 categorical levels showing the lowest degree of relative bias. Although relative bias was smaller for the self-

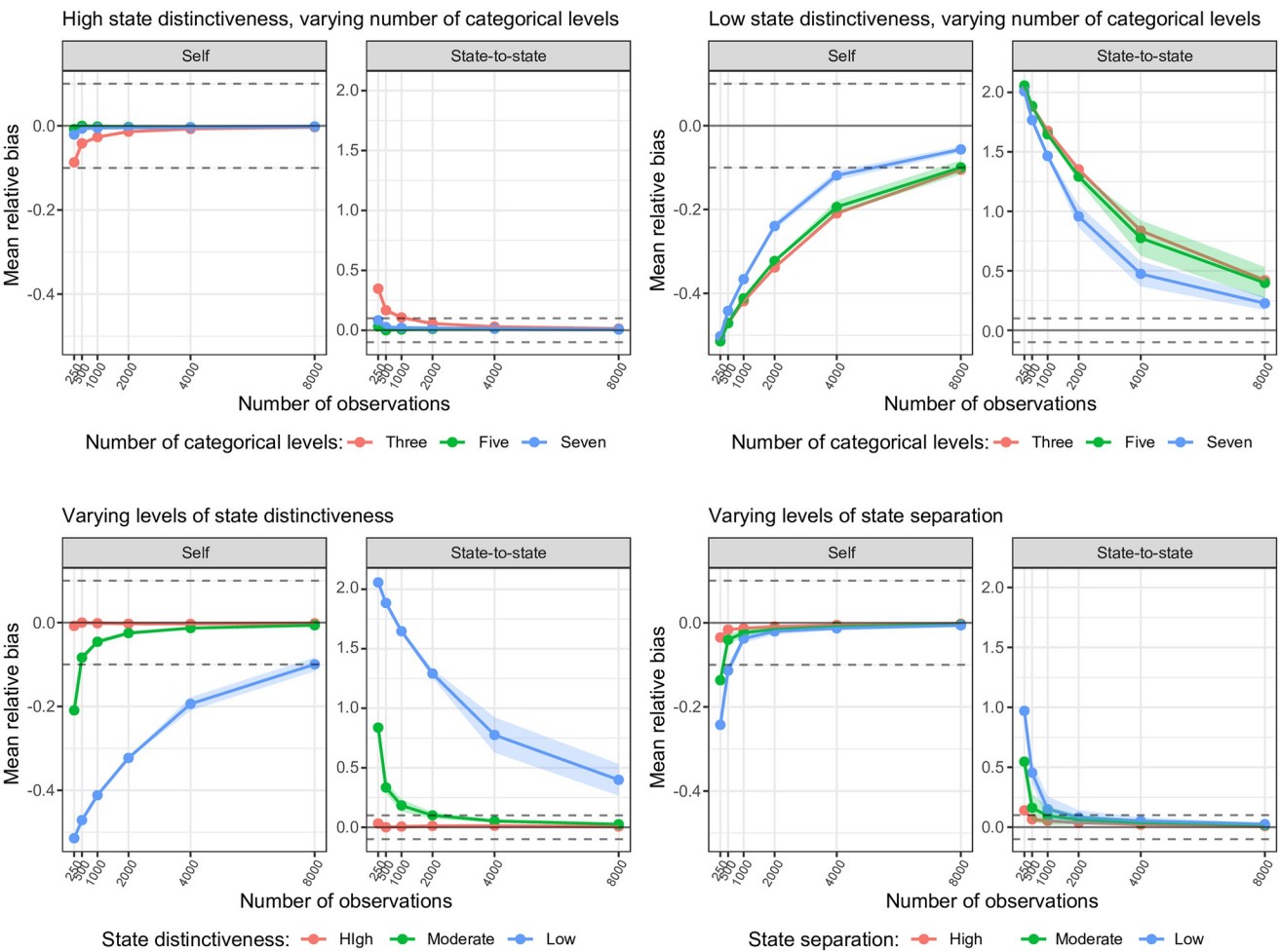

**Fig 2. Trellis plot of the mean relative bias for the estimation of the self and state-to-state transition probabilities.** Plot shows subset of scenarios with high state distinctiveness (upper left panel) and low state distinctiveness (upper right panel) over levels of number of categorical variables (line color) and number of observations, and the subset of scenarios with varying levels of state distinctiveness (bottom left panel; line color) and varying levels of state separation (bottom right panel; line color) over number of observations.

transition probabilities compared to the state-switching probabilities, the mean absolute bias was generally larger for the self-transition compared to the state-switching probabilities, see Fig A in S1 Fig.

## Precision

The level of state distinctiveness and number of observations had a large effect on the mean empirical standard error; see Figs B and E in S1 Fig for the trellis plots of the transition probabilities and emission probabilities, respectively. That is, the empirical standard error decreased with increasing number of observations, with larger decreases at the lower end of number of observations and at low state distinctiveness. In simulation scenarios with number of observations > 250 and high or moderate state distinctiveness and for all levels of state separation, the average empirical standard error of both the transition and emission probabilities were < 0.10, decreasing to < .05 when number of observations > 500. With low state distinctiveness, the empirical standard error only dove below 0.10 at 8000 observations. The number

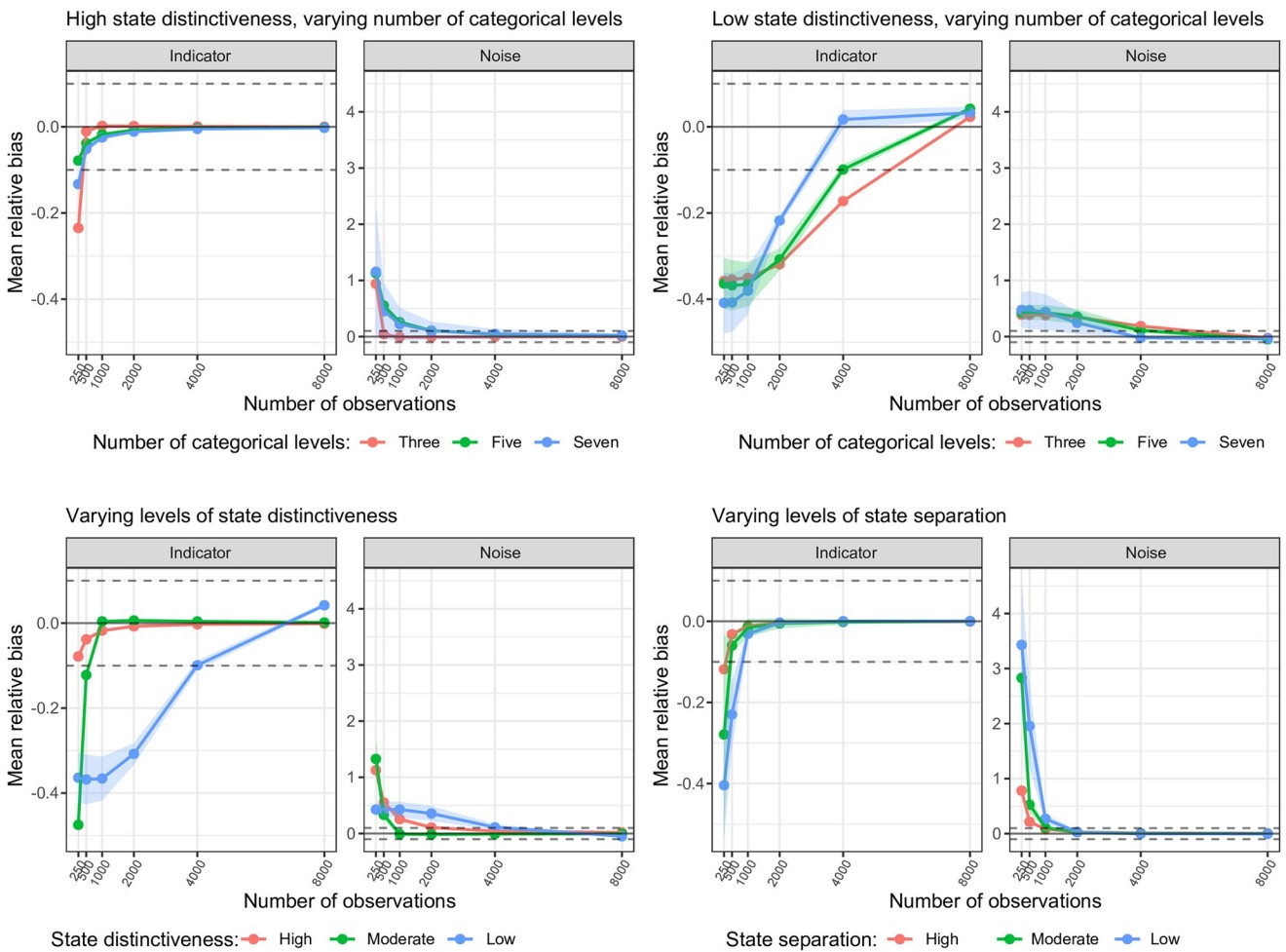

**Fig 3. Trellis plot of the mean relative bias for the estimation of the indicator and noise emission probabilities.** Plot shows the subset of scenarios with high state distinctiveness (upper left panel) and low state distinctiveness (upper right panel) over levels of number of categorical variables (line color) and number of observations, and the subset of scenarios with varying levels of state distinctiveness (bottom left panel; line color) and varying levels of state separation (bottom right panel; line color) over number of observations.

of categorical levels only had a marginal effect on the empirical standard error, with the lowest empirical standard error being observed for seven categorical levels.

## Coverage

For the transition probabilities, all simulation scenarios showed acceptable mean coverage of the 95% CrI for both the self-transition and state-switching probabilities (range: 95—100%), except for the self-transition probabilities in the low state distinctiveness scenario, see Fig 4. For the latter scenario, mean coverage increased with increasing number of observations, but was still slightly below 95% at 8000 observations. The undercoverage in the self-transition probabilities in this particular scenario was a result of the high parameter bias: the mean bias-corrected coverage showed acceptable coverage for all number of observations, see Fig C in S1 Fig.

For the emission probabilities, the difference in mean coverage of the 95% CrI was not as pronounced between levels of state distinctiveness, but in general, >500 observations were

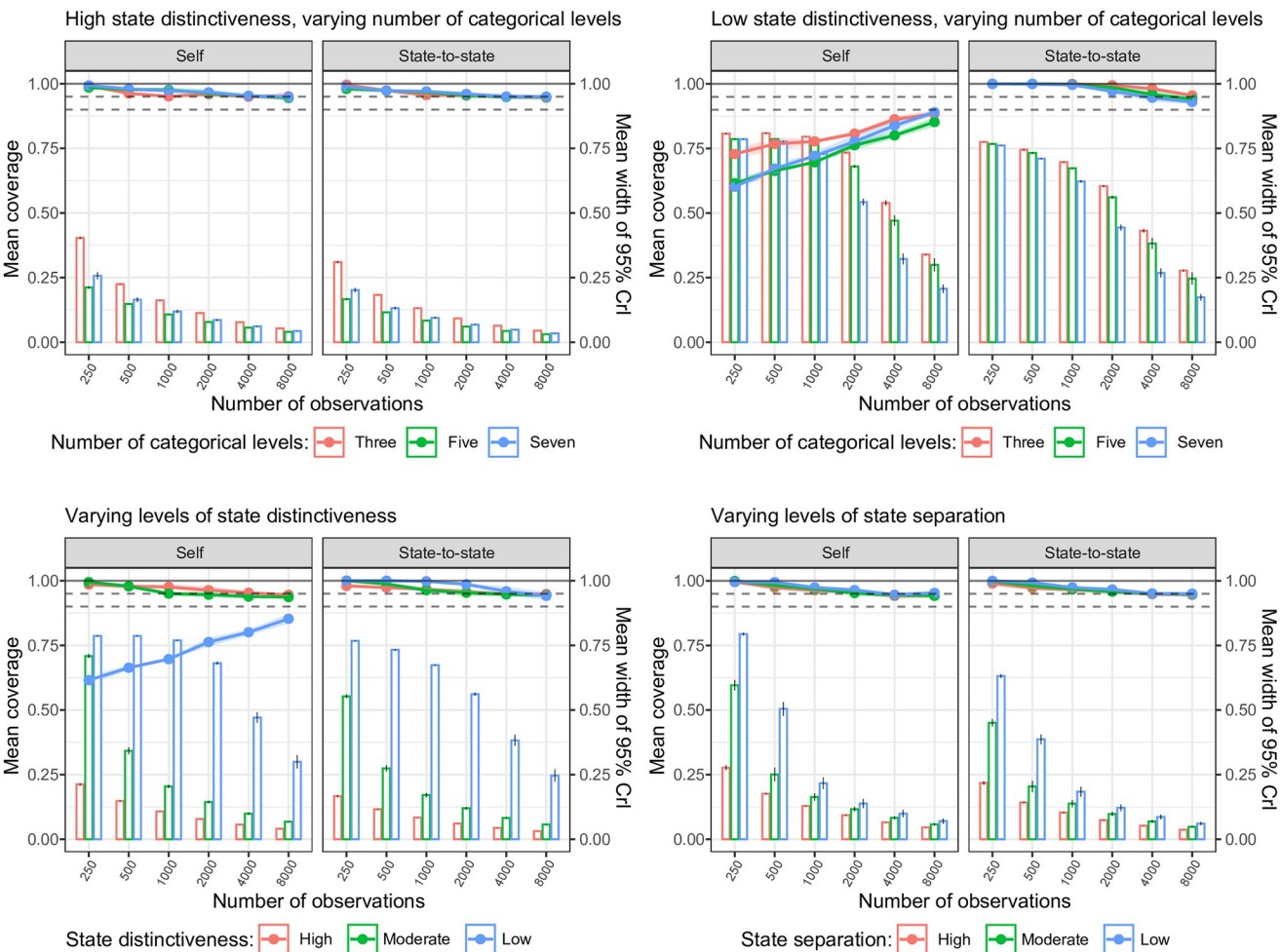

**Fig 4. Trellis plot of the mean coverage for the estimation of the self and state-to-state transition probabilities.** Plot shows the subset of scenarios with high state distinctiveness (upper left panel) and low state distinctiveness (upper right panel) over levels of number of categorical variables (line color) and number of observations, and the subset of scenarios with varying levels of state distinctiveness (bottom left panel; line color) and varying levels of state separation (bottom right panel; line color) over number of observations. Bars represent the mean width of the 95% credibility interval (CrI), right sided y-axis.

required to reach coverage levels ≥ 95%, see Fig 5. To reach a mean coverage level of 90%, 500 observations sufficed. In addition, for emission probabilities, mean coverage decreased with increasing number of observations in the low-state distinctness scenario. As the mean bias-corrected coverage in Fig F in S1 Fig showed, the undercoverage in this instance was not related to bias, but to too narrow widths of the 95% CrIs. For both transition and emission probabilities, the mean coverage did not vary by the number of categorical levels, and the width of the 95% credible interval decreased with increasing number of observations, most pronounced with higher levels of data complexity (i.e., low or moderate levels of state distinctiveness or state separation).

## Discussion

In this study, we investigated the influence of observation sequence features on the performance of parameter estimation for the Bayesian hidden Markov model (HMM). We varied

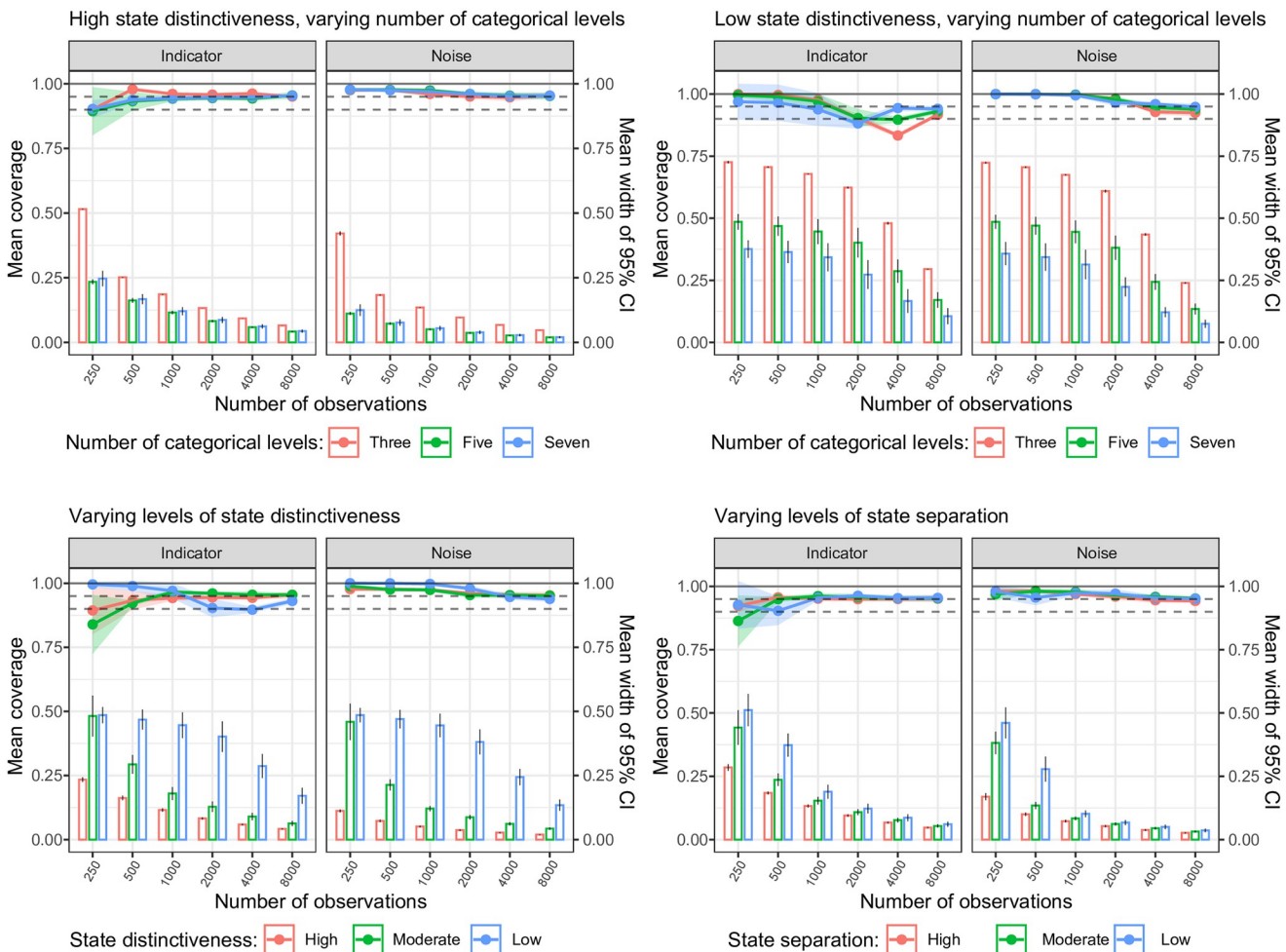

**Fig 5. Trellis plot of the mean coverage for the estimation of the indicator and noise emission probabilities.** Plot shows the subset of scenarios with high state distinctiveness (upper left panel) and low state distinctiveness (upper right panel) over levels of number of categorical variables (line color) and number of observations, and the subset of scenarios with varying levels of state distinctiveness (bottom left panel; line color) and varying levels of state separation (bottom right panel; line color) over number of observations. Bars represent the mean width of the 95% credibility interval (CrI), right sided y-axis.

the number of observations, number of categorical levels, and level of state distinctiveness and state separation. Information on what constitutes a minimum sample size to achieve adequate model performance and how the minimal sample size varies with number of categorical levels and state distinctiveness and separation is crucial to ensure reliable and replicable results when utilizing the HMM on categorical data.

## Main findings and recommendations for researchers

In general, the Bayesian HMM showed adequate levels of accuracy, precision and coverage. Model performance only minimally varied with levels of state separation and number of categorical levels. The level of state distinctiveness and number of observations, however, did have a large effect on model performance, with model performance increasing substantially with higher levels of state distinctiveness and a higher number of observations. In addition, convergence was generally achieved across all simulation scenarios, except in the case of low-state distinctiveness.

Researchers can generally expect to obtain converged, accurate, and precise estimates within the 95% credible interval for both the transition and emission probabilities from a sequence number of observations of 1.000 on-wards, assuming that the level of state distinctiveness is at least moderate. That is, to reach adequate convergence across all levels of state separation, > 500 observations were required. Convergence was not achieved when state distinctiveness was low, irrespective of the number of observations. In addition, low state distinctiveness had a detrimental effect on model performance, which could only partly be compensated by increasing the number of observations. As such, when state distinctiveness is low, the performance of the Bayesian HMM will likely be affected at even the highest number of observations. There is evidence from other latent mixture modeling approaches for categorical data, such as latent class analysis (LCA), that the detrimental effect of low state distinctiveness on model performance is general. Within the LCA literature, variables are referred to as being of high quality—i.e., variables showing a strong relationship to the latent class variable with conditional response probabilities close to one—or low quality—i.e., variables showing a weak relationship to the latent cluster, in our study equivalent to low probabilities for the indicator category and high noise probabilities. In a study examining the performance of latent class analysis, parameter bias was high in scenarios with variables of low quality, even with increased sample size [57].

An unexpected result was that seven categorical levels exhibited the lowest relative bias and empirical standard error in scenarios with low state distinctiveness, compared to three and five categorical levels. This finding is consistent with prior research suggesting that increasing the number of categorical levels can reduce auto-correlation in the observation sequence, which may, in turn, improve HMM performance. More specifically, a study on the effect of pattern structure on HMM estimation found that HMM performance improved as the number of categorical levels in the observation sequence increased [58]. Similarly, in our simulation, the increase in categorical levels may have reduced auto-correlation as well, resulting in enhanced model performance.

## Future research

In the present study, the number of latent states was fixed to a total of three throughout. However, the number of latent states is directly related to the number of model parameters that must be estimated and, as such, likely to be an important determinant of the performance of the Bayesian HMM. Implementing an extension of the simulation design which varies the number of latent states is a clear avenue for future work. In addition, the results showed that $n > 500$ is required to obtain good model performance. However, how much more is required, and whether this number is closer to 500 or to 1000 (i.e., the next level chosen within the current simulation factor) is unknown but could be important information to the applied researcher. That is, when gathering observations is costly, increasing the number of observations with 100 or 500 observations can make a substantial difference in a study's effort and cost requirements. Finally, this study is limited to uni-variate data. That is, the model is fitted using the input of one dependent variable only. However, research has shown that in the case of Bayesian multilevel HMMs, researchers are advised to prioritize multivariate data [19]. In future research, it would be interesting to see to what extent this recommendation holds for the one-level Bayesian HMM.

## Supporting information

**S1 Appendix. Starting probabilities of transition and emission probability matrices.**
(PDF)

**S1 Fig. Trellis plots of mean absolute bias, mean empirical standard error, and mean bias-corrected coverage of transition and emission probabilities.**
(PDF)

## Author Contributions

**Conceptualization:** Emmeke Aarts.

**Formal analysis:** Jan-Willem Simons, Bart-Jan Boverhof.

**Investigation:** Jan-Willem Simons, Bart-Jan Boverhof.

**Methodology:** Jan-Willem Simons, Bart-Jan Boverhof.

**Resources:** Emmeke Aarts.

**Supervision:** Emmeke Aarts.

**Visualization:** Jan-Willem Simons.

**Writing – original draft:** Jan-Willem Simons, Bart-Jan Boverhof, Emmeke Aarts.

**Writing – review & editing:** Jan-Willem Simons, Bart-Jan Boverhof, Emmeke Aarts.

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
