## [Decision Letter · Decision Letter 0]

30 Aug 2024

PONE-D-24-25081The influence of observation sequence features on the performance of the Bayesian hidden Markov model: a Monte Carlo simulation studyPLOS ONE

Dear Dr. Simons,

Thank you for submitting your manuscript to PLOS ONE. After careful consideration, we feel that it has merit but does not fully meet PLOS ONE’s publication criteria as it currently stands. Therefore, we invite you to submit a revised version of the manuscript that addresses the points raised during the review process.

Both reviewers found the paper to be innovative and recommended a minor revision. After my reading, I agree with their comments and suggest that the authors respond to the reviewers' questions one by one and make the relevant changes in the body of the text

We look forward to receiving your revised manuscript.

Kind regards,

Peida Zhan

Academic Editor

PLOS ONE

Journal Requirements:

Additional Editor Comments:

Both reviewers found the manuscript to be innovative and recommended a minor revision. After my reading, I agree with their comments and suggest that the authors respond to the reviewers' questions one by one and make the appropriate changes in the body of the text.

Reviewers' comments:

Reviewer's Responses to Questions

**Comments to the Author**

1. Is the manuscript technically sound, and do the data support the conclusions?

Reviewer #1: Yes

Reviewer #2: Yes

2. Has the statistical analysis been performed appropriately and rigorously? 

Reviewer #1: Yes

Reviewer #2: Yes

3. Have the authors made all data underlying the findings in their manuscript fully available?

Reviewer #1: Yes

Reviewer #2: Yes

4. Is the manuscript presented in an intelligible fashion and written in standard English?

Reviewer #1: Yes

Reviewer #2: Yes

5. Review Comments to the Author

Reviewer #1: The authors investigate the influence of observation sequence features on the performance of parameter estimation for the Bayesian HMM through a series of Monte Carlo simulations, and discuss the main findings. The study is of practical value for application researchers. Nonetheless, there are a few minor issues that warrant the authors’ attention, and revisions are advised to address these concerns.

1. In lines 28-30 of the manuscript, it is recommended to substantiate the claim that “In the case of uni-variate data, a minimum of 800 observations on five individuals were required to obtain adequate model performance in a data set where the states are distinctive and well separated.”

2. The label in the second line of Table 2 should be corrected to “Number of observations”.

3. The authors should not only present the simulation results but also thoroughly explain and discuss the underlying reasons for these findings, as these are typically of greater interest to readers. For instance, in lines 223-224 and 238-240, it is crucial to explore why 7 categorical levels exhibit the lowest degree of relative bias and empirical standard error in scenarios of low state distinctiveness.

4. In the lines 283-286 of the Discussion section, it is mentioned that “In addition, low state distinctiveness had a detrimental effect on model performance, which could only partly be compensated by increasing the number of observations. As such, when state distinctiveness is low, the performance of the Bayesian HMM will likely be affected at even the highest number of observations.” Given this, when state distinctiveness is low, it appears that model performance remains compromised even with a high number of observations. Could the choice of methodology for fitting the model influence these results? Specifically, the authors have employed the R package mHMMbayes for model fitting. Would using a different method potentially alter the finding that low state distinctiveness degrades model performance?

Reviewer #2: This study uses Monte Carlo simulations to evaluate the impact of the number of observations, the number of levels in the categorical outcome variable, state distinctiveness, and state separation in the emission distribution on the performance of a one-level Bayesian Hidden Markov Model. However, I have the following questions and comments:

1. Is the term "one-level data" synonymous with "uni-variate data"? If so, it would be beneficial to either use consistent terminology throughout the paper or provide a clear definition to avoid confusion.

2. Why were the emission probabilities assumed to follow a categorical distribution? Furthermore, what is the rationale behind setting the number of hidden states to three? Is there empirical evidence or theoretical justification for these choices that ensures their applicability across most scenarios?

3. Can state distinctiveness and state separation be quantitatively described? It would be helpful to have a precise definition or measure for these concepts, as they appear to be critical factors in the model's performance.

4. The scope of this study is limited to one-level Bayesian HMMs, but the abstract does not explicitly mention this limitation. To ensure the conclusions are rigorous and clearly defined, the abstract should be revised to reflect this focus accurately.

6. PLOS authors have the option to publish the peer review history of their article (what does this mean?). If published, this will include your full peer review and any attached files.

Reviewer #1: No

Reviewer #2: No

---

## [Author Response · Author response to Decision Letter 0]

16 Oct 2024

Thank you for taking the time and effort to review this manuscript. Your feedback has been invaluable in improving its quality. Please find our responses to your comments below.

Comments reviewer #1: 

1. In lines 28-30 of the manuscript, it is recommended to substantiate the claim that “In the case of uni-variate data, a minimum of 800 observations on five individuals were required to obtain adequate model performance in a data set where the states are distinctive and well separated.” 

We substantiated this claim by citing the source which reports on these findings, namely “Mildiner Moraga, S., & Aarts, E. (2024). Go multivariate: recommendations on Bayesian multilevel hidden Markov models with categorical data. Multivariate Behavioral Research, 59(1), 17-45.”.

2. The label in the second line of Table 2 should be corrected to “Number of observations”. 

We corrected the label in the second line of Table 2 from “Number of categorical levels” to “Number of observations”. 

3. The authors should not only present the simulation results but also thoroughly explain and discuss the underlying reasons for these findings, as these are typically of greater interest to readers. For instance, in lines 223-224 and 238-240, it is crucial to explore why 7 categorical levels exhibit the lowest degree of relative bias and empirical standard error in scenarios of low state distinctiveness.

We added a possible explanation for the result that seven categorical levels exhibited the lowest degree of relative bias and empirical standard error in scenarios of low state distinctiveness. To the discussion, we added: “An unexpected result was that seven categorical levels exhibited the lowest relative bias and empirical standard error in scenarios with low state distinctiveness, compared to three and five categorical levels. This finding is consistent with prior research suggesting that increasing the number of categorical levels can reduce auto-correlation in the observation sequence, which may, in turn, improve HMM performance. More specifically, a study on the effect of pattern structure on HMM estimation found that HMM performance improved as the number of categorical levels in the observation sequence increased \\cite{chudova2002pattern}. Similarly, in our simulation, the increase in categorical levels may have reduced auto-correlation as well, resulting in enhanced model performance.”

4. In the lines 283-286 of the Discussion section, it is mentioned that “In addition, low state distinctiveness had a detrimental effect on model performance, which could only partly be compensated by increasing the number of observations. As such, when state distinctiveness is low, the performance of the Bayesian HMM will likely be affected at even the highest number of observations.” Given this, when state distinctiveness is low, it appears that model performance remains compromised even with a high number of observations. Could the choice of methodology for fitting the model influence these results? Specifically, the authors have employed the R package mHMMbayes for model fitting. Would using a different method potentially alter the finding that low state distinctiveness degrades model performance?

We argue that a different method would likely not alter the finding that low state distinctiveness degrades model performance, because there is evidence from other latent mixture modeling approaches for categorical data, such as latent class analysis (LCA), that the detrimental effect of low state distinctiveness on model performance is general. To argue this point, we added the following text to the discussion: “There is evidence from other latent mixture modeling approaches for categorical data, such as latent class analysis (LCA), that the detrimental effect of low state distinctiveness on model performance is general. Within the LCA literature, variables are referred to as being of high quality - i.e., variables showing a strong relationship to the latent class variable with conditional response probabilities close to one - or low quality - i.e., variables showing a weak relationship to the latent cluster, in our study equivalent to low probabilities for the indicator category and high noise probabilities. In a study examining the performance of latent class analysis, parameter bias was high in scenarios with variables of low quality, even with increased sample size \\cite{wurpts2014}.”.

Comments reviewer #2: 

1. Is the term "one-level data" synonymous with "uni-variate data"? If so, it would be beneficial to either use consistent terminology throughout the paper or provide a clear definition to avoid confusion.

The term "one-level data" is not synonymous with "univariate data." "One-level data" refers to data collected at a single level of measurement without any nesting or hierarchical structure. In contrast, "univariate data" refers to data where only a single rather than multiple outcome variables is measured at a particular time point. We added the following sentence in the third paragraph of the introduction to clarify this distinction and avoid confusion: “We define 'one-level data' as sequence data collected at a single level of measurement for each individual, in contrast to multi-level data, where multiple sequences of data are collected for each individual and nested within these individuals in the analysis. Additionally, 'uni-variate data' refers to data where only a single outcome variable is measured at each point in time, rather than multiple outcomes.”.

2. Why were the emission probabilities assumed to follow a categorical distribution? Furthermore, what is the rationale behind setting the number of hidden states to three? Is there empirical evidence or theoretical justification for these choices that ensures their applicability across most scenarios? 

We assumed a categorical distribution for the emission probabilities because of the increasing availability of longitudinal categorical data in the social and behavioral sciences over the past decade. Our focus on examining the conditions under which Bayesian HMMs yield accurate and reliable inferences for these types of data was motivated by this development. We have included an section which elaborates on our focus and motivation, supported by relevant sources, to the introduction: “One of the key tasks of the HMM is to estimate its parameters so that they maximize the probability of an observation sequence given the HMM \\cite{rabiner1986introduction}. Obtaining a high quality solution to this optimization task is key to most applied research because it enables researchers to model real-world phenomena. The features of the observation sequence are a crucial determinant of the quality of this solution \\cite{rabiner1986introduction}. In this simulation study, we assess if and how three central features of categorical observation sequences affect the performance of the Bayesian HMM: (1) the number of observations comprising the sequence, (2) the number of levels in the categorical outcome variable, and (3) state distinctiveness and separation. Our focus on categorical longitudinal data is driven by the increasing availability of such data in the social and behavioral sciences. Technological advancements have improved the efficiency and affordability of gathering high-resolution data from individuals and groups \\cite{ariens2020time, cabrera2018matchnmingle, hamaker2017no, lemaignan2018pinsoro, orfanos2017using}. Much of this data is furthermore categorical in nature, e.g., reflect social actions like drinking, speaking, and laughing during free-standing conversations and speed dating \\cite{cabrera2018matchnmingle} or interactive behaviours in group therapies for people with schizophrenia, like asking questions, giving advice, and using humor \\cite{orfanos2017using}.” We finally acknowledge that emission probabilities that follow other distributions, such as Gaussian and Poisson, are also common and relevant, but these are beyond the scope of this study.

We added an empirical justification for fixing the number of hidden states to three in the “Data generation and model fitting” section. More specifically, we added the following segment: “To determine the amount of hidden states, we screened recent articles (published after 2000) which used either basic frequentist or Bayesian HMMs for parameter estimation \\cite{alshamaa2019hidden, habayeb2017use, jiang2019dynamic, liu2020driving, lu2019weekly, manogaran2018machine, pastell2018hidden, petersen2018modeling, putland2018hidden, soruri2017hidden, ullah2018prediction, yao2017v2x} as reported in \\cite{mor2021systematic}. This procedure revealed that most studies analyzed between 2 and 15 hidden states, with the majority focusing on the 3 to 5 range. On that basis, we generated data by setting the number of states to $m = 3$.“ 

3. Can state distinctiveness and state separation be quantitatively described? It would be helpful to have a precise definition or measure for these concepts, as they appear to be critical factors in the model's performance.

We added a quantitative description of state distinctiveness and separation under the “State distinctiveness and separation” tab in the “Methods” section using the signal to noise ratio and Kullback–Leibler divergence, respectively: “The state distinctiveness and state separation features are both assigned three levels: ``High'', ``Moderate'', and ``Low''. Within our simulation study, the categorical emission distributions are constructed such that one to three categories have a high emission probability within a given state $S$. We refer to these as the indicator categories. Non-indicator categories we refer to as noise categories. Given that all emission probabilities within a state sum to one, larger probabilities for the noise categories inevitably result in lower probabilities for the indicator categories and hence lower state distinctiveness. As such, state distinctiveness can be seen as a signal to noise ratio (SNR) of the indicator category: $SNR = P_{indicator} / P_{noise}$, with higher values representing higher state distinctiveness. Depending on the number of categorical levels and number of indicator variables within a state, the SNR ranges between 8.00 and 46.00 for High state distinctiveness, between 4.75 and 8.50 for Moderate state distinctiveness, and between 2.07 and 4.00 for Low state distinctiveness. Over the levels of state distinctiveness, the SNR values steadily increase within any given number of categorical levels. Only within the scenario of 5 categorical levels are all levels of the factor state distinctiveness applied. For the remaining levels of the factor categorical levels, only high and low state distinctiveness is used. 

The amount of state separation is quantified using the summed pairwise Kullback–Leibler divergence $D_{KL}$, with higher values denoting a larger difference, i.e., state separation. In a model with $m = 3$ states, the pairwise comparisons refer to the $D_{KL}$ of state 1 versus 2, state 1 versus 3, and state 2 versus 3, with the summed pairwise $D_{KL}$ referring to the summation over these three comparisons. In the High state separation scenario, each category is a unique indicator for one hidden state only. In the moderate state separation scenario, each hidden state is composed of one or two categories that are a unique indicator to that hidden state only, plus a category that is shared as an indicator over two states. Here, the probability of the unique indicator is higher compared to the shared indicator. In the low state separation scenario, the probabilities of the unique and shared indicators are equal. This results in a summed pairwise $D_{KL}$ for the High, Moderate, and Low state separation scenarios of 6.65, 4.44 and 3.88, respectively. Only within the scenario of 5 categorical levels, varying levels of state separation are examined. To keep this study concise, the levels of state distinctiveness and state separation are not crossed. An overview of the used emission probability matrices is provided in Table \\ref{table1}.”

4. The scope of this study is limited to one-level Bayesian HMMs, but the abstract does not explicitly mention this limitation. To ensure the conclusions are rigorous and clearly defined, the abstract should be revised to reflect this focus accurately.

We clarified in the abstract that the scope of the study is limited to the application of the Bayesian HMM to categorical, one-level data by adding the following sentence: “There is a lack of information on the estimation performance of the Bayesian hidden Markov model when applied to categorical, one-level sequence data.”

Changes to reference list

We added the following sources to the reference list to support the argument that the availability of categorical longitudinal data has increased in the social and behavioral sciences: 

Ariens, S., Ceulemans, E., & Adolf, J. K. (2020). Time series analysis of intensive longitudinal data in psychosomatic research: A methodological overview. Journal of Psychosomatic Research, 137, 110191.

Cabrera-Quiros, L., Demetriou, A., Gedik, E., van der Meij, L., & Hung, H. (2018). The MatchNMingle dataset: a novel multi-sensor resource for the analysis of social interactions and group dynamics in-the-wild during free-standing conversations and speed dates. IEEE Transactions on Affective Computing, 12(1), 113-130.

Hamaker, E. L., & Wichers, M. (2017). No time like the present: Discovering the hidden dynamics in intensive longitudinal data. Current Directions in Psychological Science, 26(1), 10-15.

Lemaignan, S., Edmunds, C. E., Senft, E., & Belpaeme, T. (2018). The PInSoRo dataset: Supporting the data-driven study of child-child and child-robot social dynamics. PloS one, 13(10), e0205999.

Orfanos, S., Akther, S. F., Abdul-Basit, M., McCabe, R., & Priebe, S. (2017). Using video-annotation software to identify interactions in group therapies for schizophrenia: assessing reliability and associations with outcomes. BMC psychiatry, 17, 1-10.

We added the following source to cite evidence from other latent mixture modeling approaches for categorical data, such as latent class analysis (LCA), that the detrimental effect of low state 

distinctiveness on model performance is general: 

Wurpts, I. C., & Geiser, C. (2014). Is adding more indicators to a latent class analysis beneficial or detrimental? Results of a Monte-Carlo study. Frontiers in psychology, 5, 920.

We finally added the following source as evidence which suggest that detecting structured patterns in a Markov context presents a more challenging learning problem than detecting more random patterns, as structured patterns violate the assumption of independent and identically distributed observations:

Chudova, D., & Smyth, P. (2002, July). Pattern discovery in sequences under a markov assumption. In Proceedings of the eighth ACM SIGKDD international conference on Knowledge discovery and data mining (pp. 153-162).

---

## [Decision Letter · Decision Letter 1]

12 Nov 2024

The Influence of Observation Sequence Features on the Performance of the Bayesian Hidden Markov Model: a Monte Carlo Simulation Study

PONE-D-24-25081R1

Dear Dr. Aarts,

We’re pleased to inform you that your manuscript has been judged scientifically suitable for publication and will be formally accepted for publication once it meets all outstanding technical requirements.

Kind regards,

Peida Zhan

Academic Editor

PLOS ONE

Additional Editor Comments (optional):

Reviewers' comments:

Reviewer's Responses to Questions

**Comments to the Author**

1. If the authors have adequately addressed your comments raised in a previous round of review and you feel that this manuscript is now acceptable for publication, you may indicate that here to bypass the “Comments to the Author” section, enter your conflict of interest statement in the “Confidential to Editor” section, and submit your "Accept" recommendation.

Reviewer #1: All comments have been addressed

2. Is the manuscript technically sound, and do the data support the conclusions?

Reviewer #1: Yes

3. Has the statistical analysis been performed appropriately and rigorously? 

Reviewer #1: Yes

4. Have the authors made all data underlying the findings in their manuscript fully available?

Reviewer #1: Yes

5. Is the manuscript presented in an intelligible fashion and written in standard English?

Reviewer #1: Yes

6. Review Comments to the Author

Reviewer #1: (No Response)

7. PLOS authors have the option to publish the peer review history of their article (what does this mean?). If published, this will include your full peer review and any attached files.

Reviewer #1: No

---

## [Editor Report · Acceptance letter]

29 Nov 2024

PONE-D-24-25081R1 

PLOS ONE

Dear Dr. Aarts, 

I'm pleased to inform you that your manuscript has been deemed suitable for publication in PLOS ONE. Congratulations! Your manuscript is now being handed over to our production team.

Kind regards, 

on behalf of

Dr. Peida Zhan 

Academic Editor

PLOS ONE